# Whole-Genome Sequencing of *Corallococcus* sp. Strain EGB Reveals the Genetic Determinants Linking Taxonomy and Predatory Behavior

**DOI:** 10.3390/genes12091421

**Published:** 2021-09-15

**Authors:** Yuqiang Zhao, Yanxin Wang, Chengyao Xia, Xu Li, Xianfeng Ye, Qiwen Fan, Yan Huang, Zhoukun Li, Cancan Zhu, Zhongli Cui

**Affiliations:** 1Key Laboratory of Agricultural Environmental Microbiology, Ministry of Agriculture and Rural Affairs, College of Life Sciences, Nanjing Agricultural University, Nanjing 210095, China; zhaoyuqiang123@126.com (Y.Z.); 17805003810@163.com (Y.W.); 2019216019@njau.edu.cn (C.X.); 2018116046@njau.edu.cn (X.L.); yxf@njau.edu.cn (X.Y.); 2019116043@njau.edu.cn (Q.F.); huangyan@njau.edu.cn (Y.H.); zkl@njau.edu.cn (Z.L.); 2Institute of Botany, Jiangsu Province and Chinese Academy of Sciences, Nanjing 210014, China; zhucancan858@163.com; 3Key Laboratory of Plant Immunity, Nanjing Agricultural University, Nanjing 210095, China

**Keywords:** myxobacter, comparative genomics, taxonomy, predatory

## Abstract

*Corallococcus* sp. strain EGB is a Gram-negative myxobacteria isolated from saline soil, and has considerable potential for the biocontrol of phytopathogenic fungi. However, the detailed mechanisms related to development and predatory behavior are unclear. To obtain a comprehensive overview of genetic features, the genome of strain EGB was sequenced, annotated, and compared with 10 other *Corallococcus* species. The strain EGB genome was assembled as a single circular chromosome of 9.4 Mb with 7916 coding genes. Phylogenomics analysis showed that strain EGB was most closely related to *Corallococcus interemptor* AB047A, and it was inferred to be a novel species within the *Corallococcus* genus. Comparative genomic analysis revealed that the pan-genome of *Corallococcus* genus was large and open. Only a small proportion of genes were specific to strain EGB, and most of them were annotated as hypothetical proteins. Subsequent analyses showed that strain EGB produced abundant extracellular enzymes such as chitinases and β-(1,3)-glucanases, and proteases to degrade the cell-wall components of phytopathogenic fungi. In addition, 35 biosynthetic gene clusters potentially coding for antimicrobial compounds were identified in the strain EGB, and the majority of them were present in the dispensable pan-genome with unexplored metabolites. Other genes related to secretion and regulation were also explored for strain EGB. This study opens new perspectives in the greater understanding of the predatory behavior of strain EGB, and facilitates a potential application in the biocontrol of fungal plant diseases in the future.

## 1. Introduction

Myxobacteria are a type of Gram-negative deltaproteobacteria that are distributed all over the world [1]. To date, hundreds of isolates of myxobacteria have been cultured from various natural sources including soil, compost, or trees. Unlike other bacteria, Myxobacteria exhibit several complex social behaviors, and one distinct trait is that their cells move by swarming or gliding on surfaces for social communication [2]. Once nutrients are depleted, fruiting bodies are born naked, followed by differentiation into myxospores. Due to their complex lifecycle, Myxobacteria produce abundant proteins to participate in signal transduction pathways, so as to coordinate cell–cell communication and regulate social motility [3]. In addition, myxobacteria prey on other microorganisms including fungi and bacteria, thus placing myxobacteria at the top of the microbial food chain [4,5]. The effective predation ability of myxobacteria is thought to rely on macromolecule degradation enzymes and secondary metabolites [6]. Myxobacteria can produce abundant secondary metabolites for potential applications as ecological weapons against living microorganisms [7]. Therefore, predatory behavior could make some myxobacteria suitable for the biological control of plant diseases. Five genera have been described within myxobacteria, and *Myxococcus* and *Corallococcus* are the two dominant genera, which possess ten and twelve species as of 2020 [8]. Among these, *Myxococcus xanthus* is the best-studied myxobacterium, and is widely used as an exemplar for the investigation of myxobacterial biology [9]. In contrast, species within the *Corallococcus* genus are poorly studied.

In our previous research, *Corallococcus* sp. strain EGB, a myxobacteria strain isolated from the soil, exhibited efficient biological control of various phytopathogenic fungi such as *Fusarium oxysporum* [10]. Several aspects, including hydrolytic activity and secondary metabolite, have been investigated to help understand its developmental and antifungal behaviors. A β-(1,3)-glucanase (lamC) derived from *Corallococcus* sp. strain EGB, displays exo-mode activity toward β-(1,3)/(1,6)-linked glucan substrates and endo-mode activity on β-(1,4)-linked glucan and xylan substrates, and shows lytic and antifungal activity against the plant pathogen *Magnaporthe oryzae* [11,12]. Another chitin hydrolase CcCti1 is identified from *Corallococcus* sp. strain EGB, and exhibits efficient antifungal activity against rice blast fungus [13]. Moreover, 32 volatile organic compounds have been identified from *Corallococcus* sp. strain EGB, and several compounds, such as isooctanol, could significantly inhibit the mycelial growth of *F. oxysporum* f. sp. *cucumerinum* and *Penicillium digitatum* [14]. These studies reveal that *Corallococcus* sp. strain EGB has potential applications in the control of plant pathogenic fungi; however, the detailed antifungal mechanism is still poorly investigated.

With the boom in high-throughput sequencing, whole-genome sequencing has been increasingly used in daily research. The availability of genome sequences not only allows for the rapid and reliable taxonomic classification of new strains, but also provides deeper insights into organismal evolution and gene functions [8]. Currently, more than 50 draft genomes are available for *Myxococcus* and *Corallococcus* strains, and the relatively larger genome size is a feature of Myxobacter [8,15]. These genomes possess several thousand proteins, many of which are hypothetical proteins. Comparative genomics analyses of organisms within the Myxococcaceae have clearly demonstrated many novel species, and *Corallococcus* genomes could group into two major clades [8,15]. Both the *Corallococcus* pan-genome and *Myxococcus*/*Pyxidicoccus* pan-genome are large and open, and *Myxococcus*/*Pyxidicoccus* spp. genomes are more diverse than those of *Corallococcus* species [8]. In both genera, a high number of biosynthetic gene clusters (BGCs) have been identified, and they are enriched in accessory pan-genomes with potential selective advantages during predation [8,15].

Here, we describe a novel *Corallococcus* genome, which displays efficient biological control of various phytopathogenic fungi. We assembled the chromosome-level genome of *Corallococcus* sp. strain EGB, and performed a comparative genome analysis with another ten species in the genus *Corallococcus*. The phylogenomics approach was used to confirm the taxonomic classification of strain EGB. Genes related to secretion, regulation, extracellular enzymes, and biosynthetic gene clusters (BGCs) were further analyzed to expand the understanding of the development and predatory behavior of strain EGB.

## 2. Materials and Methods

### 2.1. Culturing and DNA Isolation of Corallococcus sp. Strain EGB

*Corallococcus* sp. strain EGB (CCTCC No. M2012528) was previously isolated from saline soil, and was cultivated in VY/4 medium (1% yeast cells and 1% CaCl_2_, PH 7.0, *w/v*). Genomic DNA of *Corallococcus* sp. strain EGB was extracted using the methods described previously with minor modifications [16].

### 2.2. Genome Sequencing, Assembly, and Annotation

The genome of *Corallococcus* sp. strain EGB was sequenced using a hybrid strategy that combined the PacBio RS II platform and Illumina HiSeq 4000 platform at the Beijing Genomics Institute (BGI, Shenzhen, China). The PacBio subreads with lengths smaller than 1 kb were removed first, and the filtered subreads were used to perform self-correction using the Pbdagcon software, which resulted in corrected reads. After that, the Falcon tool was performed for genome assembly. To improve the accuracy of the genome assembly, GATK and SOAP tool packages were used to make single-base corrections. The complete genome sequence of *Corallococcus* sp. strain EGB was submitted to the NCBI GenBank with the accession number CP079946. Gene prediction and functional annotation were performed by the online platform Rapid Annotation using Subsystem Technology (RAST) [17]. To run the Clusters of Orthologous Groups (COG) annotation, only the best blast hit was retained for each protein [18].

### 2.3. Comparative Genomics Analysis

To conduct a comparative genomic analysis of *Corallococcus* sp. strain EGB with the other related bacterial species, the genome sequences of ten species belonging to the genus *Corallococcus* were retrieved from GenBank (Table 1). All the predicted protein sequences of 12 selected species were used to perform the all-versus-all blastp analysis with an E-value of 1 × 10^−5^ as the cut-off value. Then blast scores were delivered to the MCL Markov clustering program with an inflation parameter of 2, which resulted in rapid clustering of orthologous groups [19]. The core proteome was defined as proteins conserved in all the 11 *Corallococcus* genomes, and dispensable proteome consisted of proteins present in 2–10 *Corallococcus* genomes, while unique proteome was defined as proteins that only exist in the individual genome.

### 2.4. Phylogenetic Analysis

Whole-genome phylogeny was performed to confirm the taxonomic classification. Based on the clustering results of orthologous proteins, we randomly selected 135 single-copy genes, which were conserved in 12 analyzed species. Alignments of concatenated sequences of 135 single-copy genes were generated by the Muscle algorithm [20], and then the phylogenetic tree was constructed following the neighbor-joining algorithm with 1000 bootstrap replicates in MEGA7 [21]. The synteny of *Corallococcus* sp. strain EGB genome and other bacterial genomes was determined using Blast and Artemis Comparison Tool (ACT) [22]. The average nucleotide identity (ANI) between genomes was counted using orthoANI software [23]. The digital DNA-DNA hybridization (dDDH) values were calculated using the GGDC server with recommended formula 2 [24]. 

### 2.5. Bioinformatics Analyses

Putative secreted proteins of strain EGB were predicted using the SignalP v5 server [25]. Glycoside hydrolases were predicted by running dbCAN tool scans against the Carbohydrate-Active Enzymes (CAZy) database [26]. To identify peptidases, the protein sequences were used to perform Blast against the MEROPS database with an E-value cut-off of 1 × 10^−5^, and the produced peptidases were grouped based on peptidase family [27]. Biosynthetic gene clusters (BGCs) of each bacterial genome were predicted using the AntiSMASH v6 server [28].

## 3. Results

### 3.1. Genomic Characterization of Corallococcus sp. Strain EGB

The genome of *Corallococcus* sp. strain EGB was sequenced using a hybrid strategy that combined sequences from PacBio long reads and Illumina short reads. The genome was assembled as a single circular chromosome of 9.4 Mb with a high GC content of 70.4% (Figure 1; Table 1). In contrast to the genomes of 11 reported *Corallococcus* species, only *C. coralloides* strain DSM 2259 had a chromosome level genome that was similar to strain EGB, while the remaining 10 *Corallococcus* species only had draft genomes containing from 8 to 1491 contigs in each genome (Table 1), suggesting that strain EGB had a high-quality genome in genus *Corallococcus*. Subsequently, RAST-based annotation identified 7916 coding genes in the strain EGB genome, and 61.5% of the proteins were functionally annotated, while the remaining 38.5% of the proteins were hypothetical proteins. The nonrepetitive gene density in the strain EGB genome was 842 genes/Mb with an average gene length of 1082 bp. By contrast, the genome size and gene number of strain EGB were slightly smaller than the majority of other *Corallococcus* genomes (Table 1).

### 3.2. Strain EGB was Inferred as A Novel Species

In genus *Corallococcus*, more than ten species were reported. To establish the phylogenetic relationship of strain EGB with other related *Corallococcus* species, the phylogenetic tree was constructed using the concatenated sequences of 135 single-copy genes, which were conserved in all the analyzed genomes. The tree clearly showed that *Corallococcus* species grouped into two major clades, Group A and Group B (Figure 2A), which was consistent with previous taxonomy [15]. The strain EGB was located in the larger clade Group A, and clustered closely with *C. interemptor* AB047A (Figure 2A). Furthermore, the alignment of three chromosomal sequences of strain EGB, *C. coralloides* DSM 2259, and *M. xanthus* DK1622 was analyzed using ACT software. The results showed that strain EGB exhibited high synteny with *C. coralloides* DSM 2259; however, a relatively poor synteny with significant rearrangements was clearly observed between the genomes of strain EGB and *M. xanthus* DK1622 (Figure 2B).

The average nucleotide identity (ANI) and dDDH values between the above genomes were calculated. The results showed that ANI values ranged from 85.82% to 94.32% between the *Corallococcus* genomes. Notably, strain EGB had higher ANI values (90.79–91.51%) and dDDH values (41.2–43.7%) with *C. interemptor* AB047A, *C. exercitus* AB043B, *C. carmarthensis* CA043D, *C. aberystwythensis* AB050A, *C. exiguus* NCCRE002, and *C. coralloides* DSM 2259 (Table 2), which was consistent with the relationship obtained by the phylogenetic tree. Strains with dDDH values below 70% and ANI values below 95% were considered to be members of different species [29]. All comparisons of strain EGB and other *Corallococcus* genomes gave ANI values below 95% and dDDH values below 70% (Table 2), indicating that strain EGB was a novel species within Group A of *Corallococcus* genus.

### 3.3. Corallococcus Pan-Genome Was Large and Open

In order to explore the pan-genome of *Corallococcus* sp. strain EGB and ten other *Corallococcus* species, the predicted protein sequences of all these genomes were used to perform all-against-all Blastp analysis and subsequent ortholog clustering using MCL software. A total of 10,546 orthologous protein clusters were identified (Appendix A). Among these, 4086 clusters were found to be conserved in all the detected *Corallococcus* genomes, and were treated as the core proteome for the *Corallococcus* genus. The core proteome accounted for 68.9–74.1% of the total proteins in each genome, while 70.5% of strain EGB proteins belonged to the core proteome (Figure 3A, Appendix A). Based on the COG annotation, 10.6% of the core proteome participated in signal transduction, and 43.4% of the core proteome were responsible for essential biological functions such as transcription (category K), cell wall/membrane/envelope biogenesis (category M), amino acid transport and metabolism (category E), translation, ribosomal structure and biogenesis (category J), lipid transport and metabolism (category I), posttranslational modification (category O), energy production and conversion (category C), coenzyme transport and metabolism (category H), and carbohydrate transport and metabolism (category G) (Figure 3B). Notably, 15.9% of the core proteome could not be attributed to any known function. Despite the core proteome, around one quarter of the proteins in each *Corallococcus* genome were defined as the dispensable proteome, which shared an orthology with two or more genomes, but not in all genomes. These dispensable proteins were enriched in transcription (category K), mobilome: prophages and transposons (category X), and cell-wall/membrane/envelope biogenesis (category M) (Figure 3B). Moreover, from 157 to 495 unique proteins were found only in one genome, with no homology in other genomes (Appendix A). Of these genomes, EGB contained the largest number of unique proteins (Figure 3A), and only 18 of the 495 unique proteins could map to diverse COG categories, while the remaining unique proteins were annotated as hypothetical proteins with unknown functions, which might be related to adaptation to the specific ecological niche of strain EGB.

### 3.4. Genes Involved in Secretion

Protein secretion has not been well-studied in the *Corallococcus* genus. To explore the genomic potential in strain EGB for extracellular functions, the genome-wide identification of protein secretion systems was undertaken. The Sec and twin-arginine translocation (Tat) pathways are the most commonly used secretion systems for bacteria to transport unfolded and folded proteins across the cytoplasmic membrane [30]. In the Sec system, strain EGB contained all the key proteins except the chaperone SecB, which was dispensable (Appendix A). The Tat pathway consisted of three components: TatA, TatB, and TatC, and all of them were present in the strain EGB genome. Additionally, many other proteins are secreted through secretion systems numbered Type I through Type VI, with each system transporting specific proteins [31]. Genome mining showed that the primary proteins constituting the intact Type I and Type II secretion systems existed in the strain EGB genome (Figure 4, Appendix A). We also found two degenerated gene clusters encoding the Type III secretion system in the strain EGB (Appendix A), which was similar to that in *M. xanthus*. In contrast, the majority of proteins making up Type IV, Type V, and Type VI secretion systems were lost in strain EGB (Figure 4). An investigation of other Corallococcus genomes revealed that their distributions of the above secretion systems were consistent with strain EGB, except for the Type VI secretion system, which was present in *C. llansteffanensis*, *C. sicarius*, *C. praedator*, and *C. terminator*, whereas it was absent in the remaining *Corallococcus* species. A SignalP analysis showed that strain EGB possessed 1361 proteins for secretion via the Sec pathway, while 93 proteins were secreted by the Tat pathway (Figure 4, Appendix A). In contrast, other *Corallococcus* species contained more secreted proteins. These results suggested that strain EGB may utilize distinct secretion systems for protein transport compared to other Gram-negative bacteria.

### 3.5. Expansion of Microorganism Surface Degrading Enzymes

*Corallococcus* sp. strain EGB is known to exhibit antifungal activity against numerous phytopathogenic fungi and, as expected, it would produce various biochemically distinct extracellular enzymes such as chitinases, glucanases, and proteases to lyse other fungi. Based on the CAZy annotation, the strain EGB genome contained 62 genes encoding glycoside hydrolases (Appendix A), while other *Corallococcus* species possessed 67–80 glycoside hydrolases. We further focused on specific glycoside hydrolases known to degrade chitin and glucan, which were the main components of the fungal cell wall. Strain EGB was reported to show chitinase activity and β-(1,3)-glucanase activity. A total of 10 chitinases were identified in strain EGB, and they were divided into the two glycoside hydrolase (GH) families 18 and 19 (Appendix A). GH18 is a well-known chitinase family and strain EGB contained six GH18 chitinases, which was similar to the number in other *Corallococcus* genomes. Interestingly, we found four GH19 chitinases in strain EGB, whereas three such chitinases at most were predicted in other *Corallococcus* genomes. Phylogenetic analysis showed that one GH18 chitinase and two GH19 chitinases derived from strain EGB were species specifically expanded (Figure 5A). In addition, the strain EGB genome contained four β-(1,3)-glucanase-encoding genes belonging to the GH16 family (Appendix A), which resemble other *Corallococcus* species. Nearly all the identified chitinases and β-(1,3)-glucanases were distributed in the core proteome of the *Corallococcus* genus, suggesting that *Corallococcus* spp. had universal antifungal activity.

Protease is a large family of extracellular enzymes that participate in degrading components on the surface of the microorganisms. A total of 4.9–5.5% of the total genes were predicted to encode proteases in the *Corallococcus* genomes. The strain EGB genome contained 391 proteases, a slightly fewer than other *Corallococcus* species. The majority of identified proteases belonged to serine proteases, followed by metallo proteases (Figure 5B). These proteases of strain EGB were further classified into 70 families according to the MEROPS database (Appendix A). The three largest families, S09, S33, and S01, comprised 30% of the total proteases. We also identified an M36 metalloprotease with high similarity to *M. xanthus* MepA, which enables the extracellular digestion of proteins during predatory behavior [32].

### 3.6. Secondary Metabolite Gene Clusters Are Enriched in the Genome

*Corallococcus* sp. strain EGB and other *Corallococcus* species exhibited different antibacterial and antifungal activities, indicating that they have different or multiple biosynthetic gene clusters (BGCs) coding for antimicrobial compounds. A survey of the 11 *Corallococcus* genomes produced 654 BGCs, and each genome possessed an average of 57 BGCs. *C. llansteffanensis* CA051B contained the largest number of 97 BGCs, while strain EGB had the smallest number of 35 BGCs (Appendix A). It was noteworthy that nonribosomal peptide synthetases (NRPS) were the most prevalent BGCs, and each *Corallococcus* genome possessed an average of 25 NRPS gene clusters. There were also, on average, 10 hybrid NRPS-T1PKS gene clusters per genome. Compared to other *Corallococcus* genomes, strain EGB had fewer NRPS (9 BGCs) and NRPS-T1PKS (8 BGCs) gene clusters (Appendix A). 

Of the 654 predicted BGCs in the *Corallococcus* genomes, 119 possessed sequence similarities ≥ 50% with annotated BGCs in the antiSMASH database by KnownClusterBlast analysis, indicating that many predicted BGCs were similar to BGCs producing known metabolites. All of the 11 *Corallococcus* contained the BGCs with high similarity to the carotenoid biosynthetic gene cluster from *M. xanthus* [33], the BGCs associated with VEPE biosynthetic pathway from *M. xanthus* DK1622 [34], the BGCs homologous to the myxochelin A biosynthetic gene cluster from *Stigmatella aurantiaca* Sg a15 [35], the BGCs similar to the geosmin biosynthetic gene cluster from *Nostoc punctiforme* PCC 73102 [36], and BGCs associated with alkylpyrone-407 biosynthetic gene cluster from *Cystobacterineae bacterium* [7]. These known metabolites exhibited potent antibacterial or antifungal activities. As well as the BGCs encoding known metabolites, there was also a high number of unknown BGCs present in each *Corallococcus* genome. Of the 35 BGCs identified in the genome of strain EGB, 28 BGCs were present in part of 10 other *Corallococcus* genomes, suggesting that a vast number of BGCs were restricted to specific *Corallococcus* species. These gene clusters could be new resources for novel metabolites with antifungal or antibacterial activity, and an in-depth functional analysis was required to investigate this.

### 3.7. Genomic Evidence for Signal Transduction and Regulation

Signaling proteins regulate diverse and complex developmental and predatory behaviors of Myxobacteria. The main regulatory proteins include serine/threonine (Ser/Thr) kinases, one-component systems (OCSs), two-component systems (TCSs), and transcription factors (TFs). Based on genome mining, a large number of regulatory proteins were identified in strain EGB and other *Corallococcus* species (Table 3). The strain EGB contained 96 Ser/Thr kinases, which was smaller than other *Corallococcus* genomes. Interestingly, 22 of the 96 Ser/Thr kinases were predicted to be integral membrane proteins, indicating that they were involved in sensing external signals. TCS, consisting of a histidine kinase sensor (HK) and a response regulator (RR), is the dominant sense–response mechanism to regulate a wide array of physiological pathways. We identified 78 HKs, 139 RRs, and 36 hybrid HKs in the strain EGB. The strain EGB also possessed 60 OCS proteins, which linked environmental signals to cellular responses. In addition, strain EGB genome encoded numerous TFs, including DNA-binding transcriptional regulators and alternative sigma factors. A total of 51 response regulators, 53 sigma factors, and 156 transcriptional regulators were identified in the strain EGB, while more TFs were predicted in the majority of other *Corallococcus* genomes (Table 3).

## 4. Discussion

*Corallococcus* sp. strain EGB has a complex life-cycle, characterized by some social behaviors, particularly predatory activity. Strain EGB shows excellent biocontrol activity against various phytopathogenic fungi; however, the underlying mechanisms of these behaviors are still unclear. The popularization of genome sequencing allows us to uncover the potential mechanisms for a better understanding of adaptative evolution. In this study, we reported the complete genome sequence and functional annotation of strain EGB. Notably, the quality of the chromosome-level genome of strain EGB was significantly better than that of other *Corallococcus* genomes.

Due to the similar morphologies and behaviors of Myxobacteria, the taxonomic status of many Myxobacteria species is unclear. 16S rRNA gene is the most commonly used method for bacterial taxonomic classification [37]. *C. macrosporus* was recently reassigned to the *Myxococcus* genus. The type strains of *C. coralloides* and *C. exiguus* represented only one species, which was supported by 16S rDNA similarity [38]. Recently, the whole-genome sequence has been proved to be more reliable and have a greater discriminatory power than the 16S rRNA gene for taxonomic assignment. Nine species have been verified in *Corallococcus* using such a method [15]. Here, we applied three genome-based methods to confirm the taxonomic status of strain EGB. The phylogenetic tree based on 135 single-copy genes showed that strain EGB belonged to Group A of *Corallococcus* genus, and it was closer to *C. interemptor* AB047A. A further analysis of dDDH and ANI values revealed that strain EGB did not exhibit sufficient genomic similarity with existing *Corallococcus* species. Thus, strain EGB is supposed to be a novel species, and further phenotypic and biochemical analyses are required to support this. In general, phylogenomics has surpassed the 16S rRNA gene-based method in the field of clarifying the accurate taxonomic classification of bacteria [39].

As a predatory member, strain EGB is known to produce a vast number of extracellular enzymes to degrade the cell-wall components of phytopathogenic fungi. Accordingly, the strain EGB genome contained a repertoire of genes encoding glycoside hydrolases and proteases. Chitinase is the most commonly reported extracellular enzyme that is widely distributed in many biocontrol bacteria, such as *Pseudomonas* and *Bacillus* species, and is involved in the biocontrol activity against diver fungi. The strain EGB genome has 10 genes encoding for chitinases. Notably, one chitinase gene CcCti1 was cloned from strain EGB and exhibited obvious antifungal activity against rice blast fungus. In addition, five β-(1,3)-glucanase-encoding genes belonging to the GH6 and GH16 families were identified in the strain EGB genome. One β-(1,3)-glucanase gene lamC deriving from strain EGB showed degradation activity toward a broad range of β-linked polysaccharides, and was confirmed to inhibit the growth of *M**. oryzae* [12]. The presence of both chitinases and β-(1,3)-glucanases in the strain EGB genome supports the antifungal activity of this myxobacter to attack and degrade the cell wall of plant pathogenic fungi.

In contrast with extracellular enzymes, secondary metabolites have received more attention. Myxobacter are known to produce abundant secondary metabolites, and some of these have shown toxic activity to phytopathogenic fungi. In the *Corallococcus* genus, each genome possessed an average of 57 BGCs, and strain EGB contained the smallest number of 35 BGCs. Even so, stain EGB had a larger number of BGCs than *M. xanthus* DK1622. Within the *Corallococcus* genus, only a small proportion of BGCs were found to be conserved in all the analyzed *Coralloccus* genomes, and most BGCs were present in part of the selected species. Notably, 28 of 35 strain EGB BGCs were relatively enriched in the dispensable pan-genome, implying the presence of selective pressures for the evolutionary retention of BGCs. In contrast to previously characterized BGCs producing named metabolites, the strain EGB had five BGCs with high similarity to BGCs producing carotenoid, VEPE, myxochelin A, geosmin, and alkylpyrone-407. Some of these known metabolites have exhibited potent antimicrobial properties. Most BGCs of strain EGB were not found to be similar to annotated BGCs, indicating that there is considerable BGC diversity that remains unexplored. Such BGCs could be new resources for novel metabolites with antifungal activity, and more experiments are required for further investigation.

In summary, this study describes the genome of *Corallococcus* sp. strain EGB. Phylogenomic analysis exhibits a phylogenetic relationship and the diversity of strain EGB compared to other *Corallococcus* species. Further comparative genomic analyses provided diverse evidence that supported strain EGB as a predatory member. These results should encourage the further investigation of detailed mechanisms related to complex social behaviors, especially for predatory activity.

## Figures and Tables

**Figure 1 genes-12-01421-f001:**
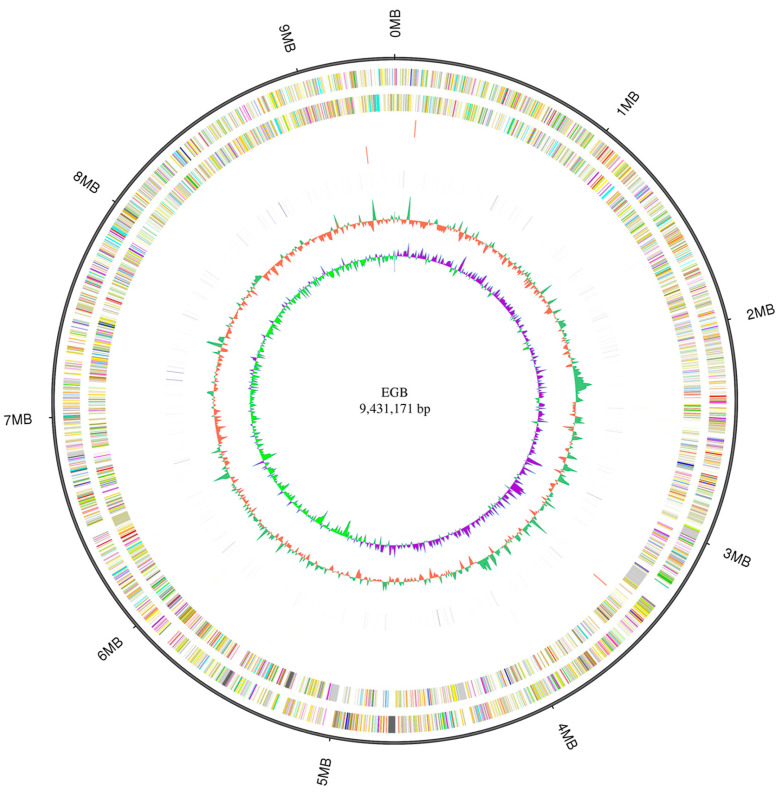
Circular representation of the complete genome of *Corallococcus* sp. strain EGB. The outermost circle is the coordinate of the genome position. From the outside to the inside, the first circle represents COG annotation gene distribution of the forwarding strand, colored according to cluster; the second circle illustrates COG annotation of the reverse strand; the third circle depicts the ncRNA distribution of the forwarding strand; the fourth circle represents the ncRNA distribution of the reverse strand; the fifth circle illustrates the repeat; the sixth circle depicts the GC content; the seventh circle denotes the GC skew.

**Figure 2 genes-12-01421-f002:**
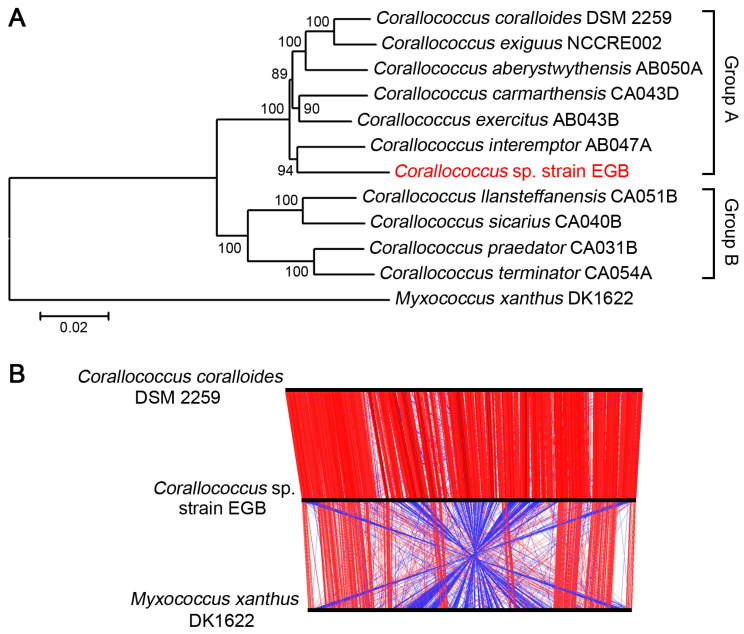
Phylogenetic relationships of strain EGB and related bacteria. (**A**) Phylogenetic tree constructed using MEGA 7 with the neighbor-joining algorithm based on concatenated sequences of 135 single-copy genes. (**B**) Linear genomic comparison of strain EGB, *C. coralloides* DSM 2259, and *M. xanthus* DK1622. The red lines in between the genomes represent matching regions, and blue lines denote inverted matching regions.

**Figure 3 genes-12-01421-f003:**
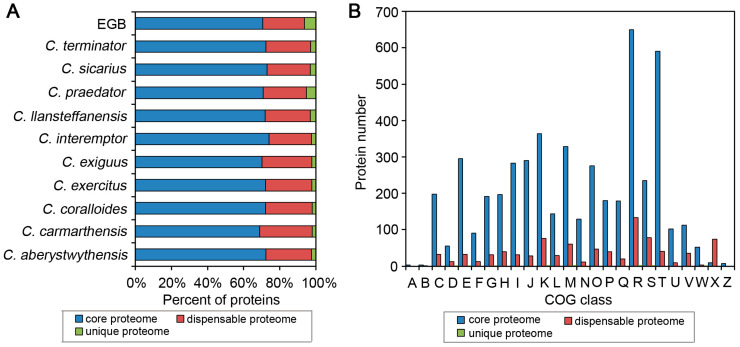
Comparative analysis of proteins between strain EGB and other *Corallococcus* species. (**A**) Number of protein families shared by different species. (**B**) Proportion of proteins enriched in the clusters of orthologous groups (COG) categories in the core, dispensable, and unique proteome. A, RNA processing and modification; B, chromatin structure and dynamics; C, energy production and conversion; D, cell-cycle control, cell division, chromosome partitioning; E, amino acid transport and metabolism; F, nucleotide transport and metabolism; G, carbohydrate transport and metabolism; H, coenzyme transport and metabolism; I, lipid transport and metabolism; J, translation, ribosomal structure and biogenesis; K, transcription; L, replication, recombination and repair; M, cell-wall/membrane/envelope biogenesis; N, cell motility; O, posttranslational modification, protein turnover, chaperones; P, inorganic ion transport and metabolism; Q, secondary metabolites biosynthesis, transport and catabolism; R, general function prediction only; S, function unknown; T, signal transduction mechanisms; U, intracellular trafficking, secretion, and vesicular transport; V, defense mechanisms; W, extracellular structures; X, mobilome: prophages, transposons; Z, cytoskeleton.

**Figure 4 genes-12-01421-f004:**
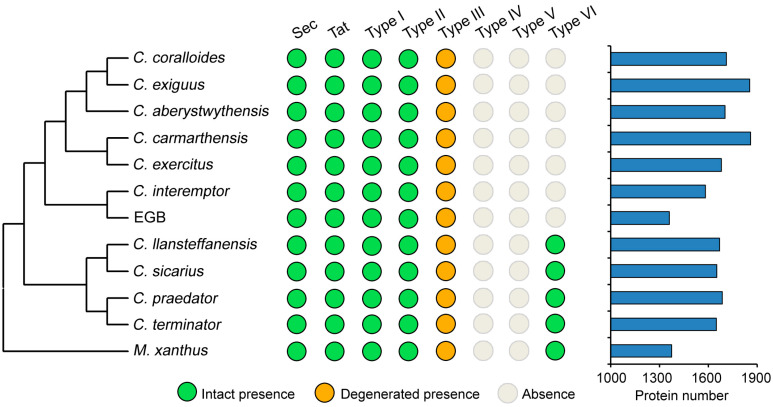
Distribution of secretion systems. The left part lists the relationship among species. The middle part shows presence or absence of each type of secretion system in each species. The right part represents the comparison of numbers of secreted proteins.

**Figure 5 genes-12-01421-f005:**
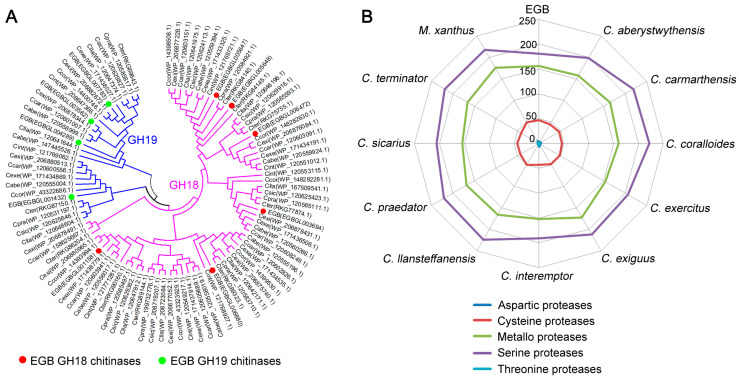
Comparison of chitinases and proteases. (**A**) Phylogenetic tree of chitinases derived from strain EGB, *C. coralloides*, *C. praedator*, and *M. xanthus*. (**B**) Comparison of protease numbers among *Corallococcus* species.

**Table 1 genes-12-01421-t001:** Genome properties of *Corallococcus* species.

Species	Strain	Genome Size (Mb)	%GC	Contigs	CDS * Number	Accession
*Corallococcus* sp.	EGB	9.4	70.4	1	7916	CP079946
*C. coralloides*	DSM 2259	10.08	69.9	1	7952	CP003389.1
*C. exiguus*	NCCRE002	10.54	69.7	8	8283	GCA_017302975.1
*C. interemptor*	AB047A	9.47	70	459	7569	GCA_003668875.1
*C. praedator*	CA031B	10.51	69.7	1491	8176	GCA_003612125.1
*C. aberystwythensis*	AB050A	9.98	70	625	8319	GCA_003612165.1
*C. llansteffanensis*	CA051B	10.53	70.3	1244	8140	GCA_003612055.1
*C. sicarius*	CA040B	10.39	70.2	802	8410	GCA_003611735.1
*C. carmarthensis*	CA043D	10.79	69.9	530	8523	GCA_003611695.1
*C. exercitus*	AB043B	10.26	70.2	690	8102	GCA_013116705.1
*C. terminator*	CA054A	10.35	69.5	863	8454	GCA_003611635.1

* CDS: Coding sequence.

**Table 2 genes-12-01421-t002:** ANI and dDDH values for pairwise comparisons between *Corallococcus* species and *M. xanthus* DK1622.

dDDH\ANI	EGB	Cint	Cexe	Ccar	Cabe	Cexi	Ccor	Clla	Csic	Cpra	Cter	Mxan
EGB	100	91.19	91.11	90.88	90.79	91.34	91.51	86.63	86.21	86.17	85.83	78.37
Cint	42.4	100	91.29	91.12	91.31	91.72	92.27	86.53	86.3	86.18	85.9	78.35
Cexe	41.9	42.7	100	92.73	92.6	91.56	91.83	87.37	87.03	86.87	86.52	78.43
Ccar	41.5	42.1	47.5	100	92.57	91.62	91.78	87.18	86.87	86.81	86.38	78.41
Cabe	41.2	42.7	47.1	47.6	100	91.55	91.84	87.04	86.73	86.7	86.2	78.41
Cexi	42.4	43.1	43.1	43.6	43.2	100	94.32	86.61	86.13	86.12	85.82	78.15
Ccor	43.7	46.1	43.9	44.4	44.3	54.1	100	86.82	86.46	86.38	85.95	78.25
Clla	30.9	30.8	32.4	32.1	31.7	30.7	31.1	100	93.42	89.27	88.82	78.78
Csic	29.9	30	31.6	31.3	31	29.8	30.2	50.4	100	88.87	88.45	78.65
Cpra	29.9	30.1	31.5	31.4	31	30	30.5	36.6	35.9	100	93.23	78.68
Cter	29.3	29.7	30.8	30.6	30.2	29.4	29.8	35.2	35	49.6	100	78.27
Mxan	21.2	21.3	21.6	21.6	21.5	21.2	21.2	21.7	21.5	21.6	21.2	100

Average nucleotide identity (ANI) values were shown above the diagonal and the digital DNA-DNA hybridization (dDDH) values were present below the diagonal. Cint, *C. interemptor* AB047A; Cexe, *C. exercitus* AB043B; Ccar, *C. carmarthensis* CA043D; Cabe, *C. aberystwythensis* AB050A; Cexi, *C. exiguus* NCCRE002; Ccor, *C. coralloides* DSM2259; Clla, *C. llansteffanensis* CA051B; Csic, *C. sicarius* CA040B; Cpra, *C. praedator* CA031B; Cter, *C. terminator* CA054A; Mxan, M. xanthus DK1622.

**Table 3 genes-12-01421-t003:** Variability in the numbers of regulatory genes per genome.

	EGB	Cabe	Ccar	Ccor	Cexe	Cexi	Cint	Clla	Cpra	Csic	Cter
Ser/Thr kinases	96	106	123	103	112	105	108	113	101	116	106
Two-component system proteins	289	299	307	303	318	297	284	322	323	312	311
Histidine kinases	146	156	157	160	167	153	151	170	162	167	163
Phosphotransfer proteins	5	5	3	2	4	2	2	5	7	4	4
Response regulators	138	138	147	141	147	142	131	147	154	141	144
Transcription factors	320	322	380	351	353	378	306	344	352	303	334
One-component system proteins	60	68	86	79	78	84	61	75	69	61	68
Response regulators	51	42	54	48	47	52	42	46	51	44	50
sigma factors	53	58	58	56	57	56	51	58	56	53	50
Transcriptional regulators	156	154	182	168	171	186	152	165	176	145	166

## Data Availability

The genome presented in this study is openly available in NCBI GenBank with the accession number CP079946.

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
