# Peer review of "Whole-Genome Sequencing of Corallococcus sp. Strain EGB Reveals the Genetic Determinants Linking Taxonomy and Predatory Behavior"

_genes, 2021, doi:10.3390/genes12091421_

Round 1
Reviewer 1 Report
In this article, the authors described the sequencing and characterisation of the genome of a Corallococcus sp isolate. They performed phylogenomics and comparative genomics and attempt to characterise protein function and link it to the predatory lifestyle of the genus.
Article is well written and, to the best of my knowledge, results support the conclusions. Just some minor comments:
-Line 36. "..isolates...have been isolated from... " it´s redundant, maybe say something like "hundreds of isolates have been cultured from" or "extracted or cultured from " to avoid repetition.
-Line 93. "was previously isolated" and remove "previously by the end of the sentence.
-Line 96. Minor instead of "slight". Also, maybe state this small modifications if believed it would be relevant to the reproducibility.
-Line 372. Change "didn´t" and "did not".
-Line 396. M. xanthus in italics.
-Line 410. "provided"
-Line 411. Remove the "help" to say "should encourage..." and add a comma after behaviors in line 413.
For future work on antimicrobials, maybe you can use the BAGEL 4 and antiSMASH tools, as they are reliable to detect potential clusters.
Author Response
Response to Reviewer 1 Comments
In this article, the authors described the sequencing and characterisation of the genome of a Corallococcus sp isolate. They performed phylogenomics and comparative genomics and attempt to characterise protein function and link it to the predatory lifestyle of the genus.
Article is well written and, to the best of my knowledge, results support the conclusions. Just some minor comments:
Response: Thanks for your positive comments.
-Line 36. "..isolates...have been isolated from... " it′s redundant, maybe say something like "hundreds of isolates have been cultured from" or "extracted or cultured from " to avoid repetition.
-Line 93. "was previously isolated" and remove "previously by the end of the sentence.
-Line 96. Minor instead of "slight". Also, maybe state this small modifications if believed it would be relevant to the reproducibility.
-Line 372. Change "didn′t" and "did not".
-Line 396. M. xanthus in italics.
-Line 410. "provided"
-Line 411. Remove the "help" to say "should encourage..." and add a comma after behaviors in line 413.
Response: All of the above English mistakes have been corrected.
For future work on antimicrobials, maybe you can use the BAGEL 4 and antiSMASH tools, as they are reliable to detect potential clusters.
Response: Thanks for your helpful suggestions. We have used antiSMASH tool in this manuscript, and we will use BAGEL 4 tool and combined their results for future work.
Reviewer 2 Report
Dear authors, for your information :
"Dear editor, thank you for asking me to review this excellent research.
I don't know if this is already a revised version but it was a pleasure to read and I have no comment at all to make.
Best regards.
Author Response
Thanks for your positive comments.